# Multisystem Inflammatory-like Syndrome in a Child Following COVID-19 mRNA Vaccination

**DOI:** 10.3390/vaccines10010043

**Published:** 2021-12-30

**Authors:** Tina Y. Poussaint, Kerri L. LaRovere, Jane W. Newburger, Janet Chou, Lise E. Nigrovic, Tanya Novak, Adrienne G. Randolph

**Affiliations:** 1Department of Radiology, Boston Children’s Hospital and Harvard Medical School, Boston, MA 02115, USA; 2Department of Neurology, Boston Children’s Hospital and Harvard Medical School, Boston, MA 02115, USA; kerri.larovere@childrens.harvard.edu; 3Department of Cardiology, Boston Children’s Hospital, Boston, MA 02115, USA; jane.newburger@cardio.chboston.org; 4Department of Pediatrics, Boston Children’s Hospital, Boston, MA 02115, USA; janet.chou@childrens.harvard.edu (J.C.); lise.nigrovic@childrens.harvard.edu (L.E.N.); adrienne.randolph@childrens.harvard.edu (A.G.R.); 5Division of Immunology, Boston Children’s Hospital, Boston, MA 02115, USA; 6Divison of Emergency Medicine, Boston Children’s Hospital, Boston, MA 02115, USA; 7Department of Anesthesiology, Critical Care and Pain Medicine, Boston Children’s Hospital and Department of Anaesthesia, Harvard Medical School, Boston, MA 02115, USA; tanya.novak@childrens.harvard.edu

**Keywords:** cytotoxic lesion of the corpus callosum, severe acute respiratory syndrome coronavirus 2, multisystem inflammatory syndrome in children, COVID-19 mRNA vaccine

## Abstract

A 12-year-old male was presented to the hospital with acute encephalopathy, headache, vomiting, diarrhea, and elevated troponin after recent COVID-19 vaccination. Two days prior to admission and before symptom onset, he received the second dose of the Pfizer-BioNTech COVID-19 vaccine. Symptoms developed within 24 h with worsening neurologic symptoms, necessitating admission to the pediatric intensive care unit. Brain magnetic resonance imaging within 16 h of admission revealed a cytotoxic splenial lesion of the corpus callosum (CLOCC). Nineteen days prior to admission, he developed erythema migrans, and completed an amoxicillin treatment course for clinical Lyme disease. However, Lyme antibody titers were negative on admission and nine days later, making active Lyme disease an unlikely explanation for his presentation to hospital. An extensive workup for other etiologies on cerebrospinal fluid and blood samples was negative, including infectious and autoimmune causes and known immune deficiencies. Three weeks after hospital discharge, all of his symptoms had dissipated, and he had a normal neurologic exam. Our report highlights a potential role of mRNA vaccine-induced immunity leading to MIS-C-like symptoms with cardiac involvement and a CLOCC in a recently vaccinated child and the complexity of establishing a causal association with vaccination. The child recovered without receipt of immune modulatory treatment.

## 1. Introduction

Encephalopathy with cytotoxic lesions of the corpus callosum (CLOCC) is increasingly being recognized in association with COVID-19 in children and adolescents [1,2,3]. CLOCCs are non-specific, non-enhancing areas of reduced diffusivity, indicating cytotoxic edema of non-vascular origin that have been observed in many pediatric infectious and non-infectious diseases [4,5,6]. CLOCCs have also been detected following mumps vaccination, which is an attenuated live virus [7,8]. 

Multisystem inflammatory syndrome in children (MIS-C) is a severe hyperinflammatory syndrome that temporally follows SARS-CoV-2 infection by 3–6 weeks and is presumably post-infectious [9]. As of 1 November 2021, there have been 5973 US cases of MIS-C associated with SARS-CoV-2 and 52 MIS-C-related deaths [10]. The Brighton Collaboration Working group recently proposed a case definition for MIS-C following COVID-19 vaccination [11]. MIS-C-like symptoms following COVID-19 mRNA vaccination have been reported in pediatric patients [12,13], neither had neurologic involvement. We conducted an extensive evaluation of a 12-year-old boy who developed neurologic and cardiac involvement following his second COVID-19 vaccination.

## 2. Case Report

We report a MIS-C-like illness in a 12-year-old boy naïve to severe acute respiratory syndrome coronavirus 2 (SARS-CoV-2) who developed encephalopathy and myocarditis following COVID-19 mRNA vaccination and was found to have an isolated CLOCC. The child lives in an endemic area for Lyme disease. Nineteen days prior to presentation (five days after his first COVID-19 vaccine), he developed a lesion consistent with erythema migrans (EM), and completed amoxicillin treatment for a clinical diagnosis of Lyme disease. He received the second dose of the Pfizer-BioNTech COVID-19 vaccine two days prior to hospital presentation. That night, he developed severe headache, and over the next 48 h had persistent headache, emesis, visual hallucinations, worsening encephalopathy and was noted to have an elevated troponin indicating cardiac involvement. A timeline of the development of MIS-C-like symptoms after Lyme disease and COVID-19 vaccination is shown in Figure 1. He was admitted to the pediatric intensive care unit for close neurologic and cardiac monitoring and underwent brain magnetic resonance imaging (MRI).

Pertinent vital signs and laboratory studies during admission were: temperature 38.3 degrees Centigrade, hyponatremia (125 mmol/L), elevated C-reactive protein (5.8 mg/dL), elevated troponin T (0.22 ng/mL), neutrophilia (9840, normal 4520–9170 cells/μL) and lymphopenia (1080, normal 1490–3110 cells/μL). He had no immunologic markers of acute or recent infection, as evidenced by no elevation of effector/memory populations within CD4^+^ or CD8^+^ T cells and a normal level of soluble CD25, a marker of T-cell activation that is increased in viral, bacterial, and tick-borne infections. On the second hospital day, BNP peaked at 190 pg/mL (100 pg/mL upper limit of normal). Neurologic examination on admission was focal with a positive Babinski response on the left. Cerebrospinal fluid (CSF) analysis was not suggestive of encephalitis. Brain MRI showed T2 prolongation and reduced diffusivity in the splenium of the corpus callosum in Figure 2. EEG showed intermittent right posterior slowing without epileptiform activity to suggest an underlying diagnosis of epilepsy. No specific immunomodulatory therapies were administered to our patient because of normal cardiac MRI and reassuring clinical trajectory and improving cardiac and inflammatory biomarkers. He was discharged on hospital day five. Diagnostic evaluation results are summarized in Table 1. Three weeks post discharge, he was asymptomatic with a normal neurologic exam. Whole-genome sequencing did not identify any coding variants indicative of an underlying immune disorder.

On admission, COVID-19 IgG (anti-SARS-CoV-2 spike) index by semi-quantitative chemiluminescent assay (ARUP Laboratories, SLC, UT) was elevated (19.83 IV, reference interval <= 0.99) whereas SARS-CoV-2 anti-nucleocapsid antibodies and respiratory swab PCR were negative, suggesting that antibodies were from vaccination and not infection. First tier Whole Cell Sonicate Lyme enzyme immunoassay (EIA) index values at time of admission and 9 days after admission were negative (0.24 acute and 0.70 convalescent).

## 3. Discussion

We present the first published report of a complex multisystem inflammatory disorder with cardiac and neurologic involvement in a 12-year-old boy naïve to SARS-CoV2 associated with the second dose of the Pfizer BioNTech vaccine. Although he had recent clinical Lyme disease, we do not believe it explains his MIS-C-like syndrome. Our patient’s clinical presentation has similarities with previous reports of CLOCCs in children in the setting of COVID-19 or MIS-C [1], probable myocarditis after COVID-19 vaccination in children [14,15,16], and CLOCC in a previously healthy young male after initial COVID-19 mRNA vaccination [17]. 

Clinical features of children with CLOCCs in association with SARS-CoV-2 and other infectious diseases typically include fever, altered consciousness (acute encephalopathy, delirium or coma), visual hallucinations, hyponatremia, and seizures [5]. Cerebrospinal fluid analysis may be normal or reveal mild pleocytosis [7,18,19,20]. Magnetic resonance imaging shows T2 prolongation in the splenium and/or genu of the corpus callosum with reduced diffusivity with or without white matter abnormalities [1]. Most patients with CLOCCs secondary to acute infections are either asymptomatic [21], or have transient neurologic symptoms with complete recovery within a few weeks with or without immunotherapies [22,23]. The pathogenesis of encephalopathy with CLOCCs remains unclear. CLOCCs are thought to be associated with increased density of cytokine and glutamate receptors in the corpus callosum, particularly the splenium, leading to cytokinopathy, increased glutamate in the extracellular space, and cytotoxic edema in the corpus callosum [1,4,24,25]. This has been corroborated by measures of elevated IL-6 and IL-10 in CSF in children with splenial lesions [19,26,27]. 

Following the FDA expansion of the Emergency Use Authorization of 2 Pfizer-BioNTech mRNA COVID-19 vaccines to include adolescents 12 to 15 years of age in May 2021, cases of post COVID-19 vaccination myocarditis and pericarditis have been reported to the U.S. Centers for Disease Control and Federal Drug Administration in this age group [28,29]. In addition to myocarditis in our patient, the diagnosis of probable post-vaccine induced MIS-C may be supported by the following features according to the Brighton criteria (Level 2b) [11]: fever for at least one day prior to admission; acute neurologic and gastrointestinal symptoms; laboratory markers of inflammation (CRP elevated); neutrophilia, lymphopenia; and elevated cardiac biomarkers (troponin and BNP levels). In the present case, SARS-CoV-2 PCR and nucleocapsid antibodies were negative, highlighting lack of SARS-CoV-2 exposure. Our patient with encephalopathy and focal neurological signs on admission, focal EEG slowing and normal CSF analysis is similar to recently reported cases of acute encephalitis in the setting of MIS-C [30].

Toll-like receptors, which are activated by mRNA vaccines, cause significant changes in cytokine and gene expression beginning within hours after receptor activation, peak within 6 to 24 h, and subside over the following 48 to 168 h [31], consistent with this patient’s disease timeline. It is unlikely that *Borrelia burgdorferi* infection alone caused the MIS-like illness. The patient’s Lyme disease diagnosis was based on the history of an EM lesion with negative acute and convalescent serology. In endemic areas, the rash alone is diagnostic of Lyme disease [32] as the sensitivity of serology is low with cutaneous disease [33,34,35] and patients may remain seronegative after appropriate treatment [36]. However, infectious pathogens may cause reversible, epigenetic reprogramming of the innate immune response prompting an atypical response to a subsequent immune challenge [37]. 

## 4. Conclusions

Our report highlights a potential role of vaccine-induced immunity leading to MIS-C-like symptoms with cardiac involvement and CLOCC in a recently vaccinated child who had an antecedent Lyme infection. Given the rarity and atypical nature of his clinical presentation, his symptoms likely arose from a complex interplay amongst multiple factors: his genetic background, recent Lyme infection, and the second dose of the COVID-19 mRNA vaccine. Importantly, the self-limited and brief course of our patient’s symptoms contrast with the sequelae of acute COVID-19, MIS-C, and post-acute sequelae of SARS-CoV-2 infection. Thus, while this case report of a rare complication expands knowledge of possible vaccine-associated events in children with recent Lyme disease, in isolation it should not be interpreted as a contraindication to SARS-CoV-2 vaccination in children.

## Figures and Tables

**Figure 1 vaccines-10-00043-f001:**
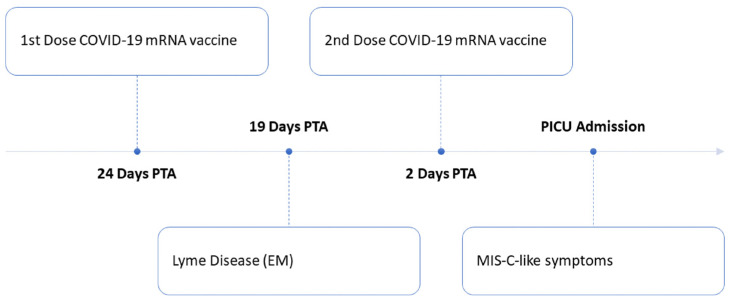
Timeline of development of MIS-C-like symptoms after Lyme disease and COVID-19 mRNA vaccination. Abbreviations: EM, erythema migrans; PICU, pediatric intensive care unit; PTA, prior to admission.

**Figure 2 vaccines-10-00043-f002:**
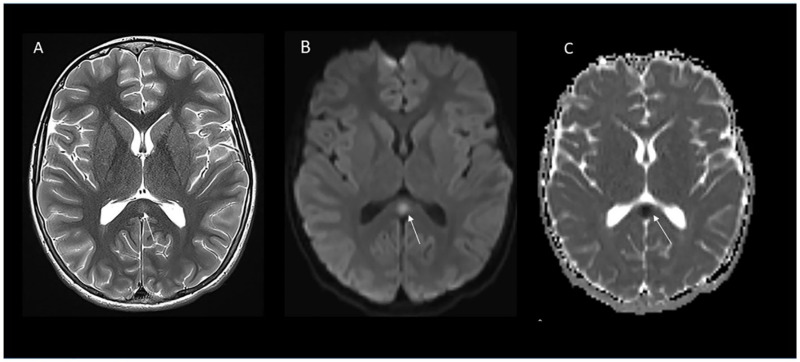
Cytotoxic lesion of the corpus callosum. Axial T2 image (**A**) shows focus of T2 prolongation representing cytotoxic edema in the splenium of corpus callosum (arrow) with reduced diffusivity on trace diffusion image (**B**) (arrow) and apparent diffusion coefficient map (**C**).

**Table 1 vaccines-10-00043-t001:** Diagnostic studies and results during hospital admission.

Laboratory Studies	Source	Result (Normal Range)
**Infectious Disease**		
SARS-CoV2 PCR	NP swab	Negative
Anti-SARS-CoV2 nucleocapsid antibodies	Blood	Negative
Culture and gram stain	CSF	Negative
Adenovirus PCR	Blood, NP swab	Negative
Enterovirus PCR	Blood, CSF	Negative
Herpes Simplex Virus type 1 and type 2	CSF	Negative
Varicella Zoster Virus PCR	CSF	Negative
Epstein Barr Virus PCR	Blood	Negative
Coxsackie A9 antibodies	Blood	Negative
Ehrlichia and Anaplasma PCR	Blood	Negative
Rickettsia rickettsia serologies, IgM, IgG	Blood	Negative
Eastern Equine Encephalitis IgM	Blood	Negative
West Nile Virus IgM	Blood	Negative
Pneumococcus IgG	Blood	Negative
Tetanus IgG	Blood	Negative
RVP (Adenovirus PCR, hMPV PCR, Rhinovirus PCR, Influenza A and B PCR, RSV PCR)	NP swab	Negative
Legionella antigen	Urine	Negative
Lyme antibody index	CSF	Negative
Neurologic		
Anti-MOG antibodies	Blood, CSF	Negative
Autoimmune encephalitis panel	Blood, CSF	Negative
CNS demyelinating disease panel ^#^	Blood	Negative
**Cardiac**		
Viral Respiratory Panel (Myocarditis)	NP Swab	Negative
Electrocardiogram (ECG)	N/A	Widespread repolarization abnormalities, with nonspecific ST-T wave changes
Echocardiogram	N/A	Normal systolic and diastolic ventricular function, normal coronary dimensions, no significant valvar dysfunction, and no pericardial effusion
Cardiac MRI	N/A	No evidence of active myocarditis or late gadolinium enhancement
**Immunology ***		
Soluble CD25, units/mL	Blood	690 (137–838)
Serum inflammatory markers ^^,^*		
D-dimer, mcg/mL FEU	Blood	0.43
Erythrocyte sedimentation rate, mm/h	Blood	5
Platelet count, K cells/μL	Blood	313
**Other ***		
White blood, cells/mm^3^	CSF	2 with 7% neutrophils, 35% lymphocytes, 58% macrophages
Red blood cells	CSF	0
Protein, mg/dL	CSF	14.9
Glucose, mg/dL	CSF	85
Opening pressure, cmH_2_O	CSF	21
Prothrombin time, seconds	Blood	15.7 (12.1–14.6)
AST, unit/L	Blood	24 (2–40)
ALT, unit/L	Blood	35 (3–30)

Abbreviations: SARS-CoV-2, severe acute respiratory syndrome coronavirus 2; NP, nasopharyngeal; CSF, cerebrospinal fluid; PCR, polymerase chain reaction; RVP, respiratory viral panel; hMPV, human metapneumovirus; RSV, respiratory syncytial virus; MOG, myelin oligodendrocyte glycoprotein; N/A, not applicable. ^#^ Includes antibodies to NMO/AQP4 (Neuromyelitis Optica/aquaporin-4) and MOG, sent 3 weeks after hospital discharge. * Maximum values during hospital admission. ^ Serum levels of procalcitonin, ferritin, lactate dehydrogenase, fibrinogen and IL6 and CSF IL-6 and IL-10 were not measured.

## Data Availability

All available supportive data are reported in this case report.

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
