# Peer review of "Multisystem Inflammatory-like Syndrome in a Child Following COVID-19 mRNA Vaccination"

_vaccines, 2021, doi:10.3390/vaccines10010043_

Round 1

Reviewer 1 Report

This is article is particularly interesting because it deals with paediatric complications following vaccination and also because it deals with the question of CNS involvement in children with MISC. These two aspects, could actually be highlighted more clearly in the article.

In fact, while it is well known that MISC is the main complication of COVID in children, there are very few reports of MISC secondary to the vaccine, and it therefore remains to be clarified whether these cases are similar to or different from the post-COVID (post-infectious) forms. 

With regard to the latter, the literature reports cases of encephalitis in the course of MISC (Olivotto, Eur J Paediatr Neurol. 2021) in which the clinical picture is very similar to that observed in this patient, but to date these aspects are poorly described in the literature. It might be interesting to include a comparison between these reported cases and the authors’ own patient.

Introduction:

It would be useful to define MISC secondary to the primary infection and to report its incidence in children.

Case report

Was an EEG performed? If so, it would be very valuable to describe it in detail and possibly compare it with what is described in the literature (Olivotto).

Author Response

Thank you for your thoughtful review. Here are our responses to Reviewer 1:

This is article is particularly interesting because it deals with paediatric complications following vaccination and also because it deals with the question of CNS involvement in children with MISC. These two aspects, could actually be highlighted more clearly in the article.

In fact, while it is well known that MISC is the main complication of COVID in children, there are very few reports of MISC secondary to the vaccine, and it therefore remains to be clarified whether these cases are similar to or different from the post-COVID (post-infectious) forms.

With regard to the latter, the literature reports cases of encephalitis in the course of MISC (Olivotto, Eur J Paediatr Neurol. 2021) in which the clinical picture is very similar to that observed in this patient, but to date these aspects are poorly described in the literature. It might be interesting to include a comparison between these reported cases and the authors’ own patient. Added to discussion.

Introduction:

It would be useful to define MISC secondary to the primary infection and to report its incidence in children. We have added the definition and incidence to the introduction.

Case report

Was an EEG performed? If so, it would be very valuable to describe it in detail and possibly compare it with what is described in the literature (Olivotto). We have added the EEG results to lines 153-155.

Reviewer 2 Report

This report of a MIS-C-like syndrome following COVID-19 mRNA vaccination is a rare case of great interest currently. His recent case of Lyme disease that occurred between the first and second dose of vaccination is intriguing and it is unclear what if any role it played.

I would suggest the following revisions to the manuscript:

  1. I would include a timeline figure of each dose of vaccine and the onset of Lyme disease and the onset of MIS-C-like symptoms.
  2. I would add that the child lived in an endemic area for Lyme disease if that is the case since an erythema migrans-like rash in a non-endemic region would be less likely to be true Lyme disease. .
  3. I would add a reference and a statement that early treatment of Lyme disease/erythema migrans frequently aborts the antibody response which would be consistent with the child's negative Lyme EIA.
  4. In the introduction, I would change the wording of the sentence with references 10 and 11. I would reword the sentence to state "MIS-C-like symptoms following COVID-19 mRNA vaccination have been reported in pediatric patients." There are other pediatric patients that have developed MIS-C-like symptoms following vaccination that have been reported to VAERS and potentially could be published prior to this case report.

Author Response

Thank you for your thoughtful review. Here are our responses to Reviewer 2.

This report of a MIS-C-like syndrome following COVID-19 mRNA vaccination is a rare case of great interest currently. His recent case of Lyme disease that occurred between the first and second dose of vaccination is intriguing and it is unclear what if any role it played.

I would suggest the following revisions to the manuscript:

I would include a timeline figure of each dose of vaccine and the onset of Lyme disease and the onset of MIS-C-like symptoms. We have added a timeline figure which is Figure 1.

I would add that the child lived in an endemic area for Lyme disease if that is the case since an erythema migrans-like rash in a non-endemic region would be less likely to be true Lyme disease. We have added a sentence to lines 65-66:"The child lives in an endemic area for Lyme disease."

I would add a reference and a statement that early treatment of Lyme disease/erythema migrans frequently aborts the antibody response which would be consistent with the child's negative Lyme EIA. A sentence has been added on lines 160-164 with references.

In the introduction, I would change the wording of the sentence with references 10 and 11. I would reword the sentence to state "MIS-C-like symptoms following COVID-19 mRNA vaccination have been reported in pediatric patients." There are other pediatric patients that have developed MIS-C-like symptoms following vaccination that have been reported to VAERS and potentially could be published prior to this case report. The sentence has been edited as suggested.